# Conveyor: Efficient Tool-aware LLM Serving with Tool Partial Execution

## Abstract

The complexity of large language model (LLM) serving workloads has substantially increased due to the integration with external tool invocations, such as ChatGPT plugins. In this paper, we identify a new opportunity for efficient LLM serving for requests that trigger tools: tool partial execution alongside LLM decoding. To this end, we design Conveyor, an efficient LLM serving system optimized for handling requests involving external tools. We introduce a novel interface for tool developers to expose partial execution opportunities to the LLM serving system and a request scheduler that facilitates partial tool execution. Our results demonstrate that tool partial execution can reduce request completion latency by up to 38.8%.

## 1 Introduction

The rapid evolution of large language models (LLMs) has significantly accelerated in recent years, and LLMs have quickly become the state-of-the-art approach in many AI tasks, such as content generation, question answering, and text classification. Consequently, LLM serving systems have emerged as a crucial component in deploying these models in various applications to achieve high performance and resource efficiency. Many LLM serving techniques have been proposed to reduce response latency Leviathan et al. (2023); Chen et al. (2023a); Fu et al. (2024); Cai et al. (2024) and improve system throughput Rasley et al. (2020); Kwon et al. (2023); Yu et al. (2022); Dao et al. (2022); Dao (2024); Chen et al. (2023b).

Recently, a new use case, *tool-assisted LLM serving*, has emerged to enhance the reasoning capabilities of LLMs and enable them to interact with the external world. In a typical tool-assisted LLM serving workflow, a user first sends a request (*i.e.*, the original prompt) to the system. An LLM will then process this request and generate a set of tool calling commands, or *plans*. Tool executors will invoke various tools to execute these commands and collect outputs, or *observations*. The original prompt, plans, and observations will be concatenated following a template, and then sent back to the LLM. The LLM generates either new plans, indicating a new round of tool execution, or the ultimate response to be sent back to the user. Example tools include but are not limited to calculators, databases, code interpreters, search engines, and ticket-booking travel agency websites. The most notable example is ChatGPT plugins, which have allowed users to book air tickets and to reason about mathematical expressions.

In this work, we identify a novel opportunity to enhance the efficiency of LLM serving systems that involve external tools. Traditional approaches to LLM serving treat the invocation of external tools as separate, sequential processes, leading to increased request completion times. However, we propose that these processes can be optimized through partial execution of tools concurrently with LLM decoding for a wide range of external tools (e.g., code interpreter, search, validation). We call this *tool partial execution*. Figure 1 shows a conceptual example of LLM serving to demonstrate the performance benefits of tool partial execution. For instance, when LLM generates Python code for data visualization, one line of code such as "`import matplotlib.pyplot as plt`" can immediately be executed in the Python interpreter before the subsequent Python code is decoded (or generated) by the LLM. With partial execution, the resulting latency can be much shorter, because the tool execution (e.g., loading Matplotlib) and LLM decoding (e.g., generating subsequent Python code) can run in parallel without blocking each other.

To this end, we build Conveyor, an LLM serving system optimized for requests that trigger external tools. Conveyor consists of two key design points. First, *we propose an interface for a tool developer*

Figure 1: An example of tool-assisted LLM serving scenarios with and without tool partial execution optimization. This example includes three rounds of LLM inference (blue and green blocks) and two rounds of tool invocation (gray blocks).

*to express the partial execution opportunity for an LLM serving system*. For example, a code interpreter can use "\n" (newline) or ";" to serve as indicators of the opportunity for tool partial execution. Second, *we build a token-granularity scheduler to detect such partial execution opportunities and invoke the corresponding tools to minimize unnecessary blocking and improve performance*. During LLM decoding, Conveyor detects the indication for partial execution opportunities, invokes tools, and collects tool invocation results for future prefilling. Conveyor is fully compatible with state-of-the-art efficient LLM serving techniques, such as PagedAttention Kwon et al. (2023), FlashAttention Dao et al. (2022); Dao (2024), and continuous batching Yu et al. (2022).

To evaluate Conveyor, we use four LLM serving workloads that contain external tool invocations and use Mistral-7B-Instruct-v0.2 Jiang et al. (2023) and Functionary-Small-v2.2[1] to invoke tools. Our evaluation focuses on two major aspects. First, we demonstrate the potentials of *tool partial execution* with Conveyor by showcasing the performance benefits across various workloads. For instance, Conveyor reduces the latency by up to 38.8% across workloads including code generation, search, and planning. Second, we intentionally explore the limitations of Conveyor. Since the performance improvements depend heavily on the characteristics of workloads and the external tools, we combine theoretical analysis with practical workload testing to thoroughly study scenarios (e.g., invoking calculator tools) where Conveyor provide only limited improvements. This two-fold evaluation allows us to present a comprehensive understanding of both the strengths and boundaries of Conveyor. We would also like to highlight that the effectiveness of Conveyor is not affected by the choice of LLM or prompts because the execution flow of LLM decoding instructions and invoking tools remains the same that tool partial execution can happen alongside LLM decoding.

In summary, this paper makes the following main contributions:

- We are the first to identify the opportunity for tool partial execution during LLM decoding;
- We build Conveyor, an LLM serving system that enables tool partial execution to significantly reduce the total request completion latency;
- We conduct systematic empirical evaluation and analysis to demonstrate that tool partial execution can provide performance benefits to a wide range of external tools.

## 2 RELATED WORK

In this section, we first introduce how modern LLM serving systems work. We next summarize emerging efforts in integrating external tool access into LLM serving.

### 2.1 LLM SERVING SYSTEMS

Modern LLMs predominantly employ the Transformer architecture Vaswani et al. (2017), at the core of which lies the *self-attention* module. The self-attention module computes three vectors for each token in the sequence, including query (Q), key (K), and value (V) vectors. It then calculates the attention score for each token by multiplying its Q vector with the K vectors of all preceding tokens, followed by a softmax and weighted average computation. Since the key and value vectors for processed tokens will be reused when generating new tokens, previous keys and values are usually cached in the GPU memory, known as the KV cache.

---

[1] https://github.com/MeetKai/functionary

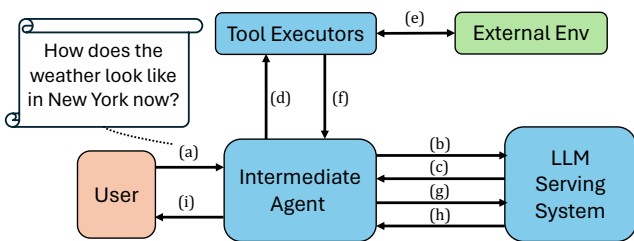

Figure 2: A tool-assisted LLM serving scenario.

To process an LLM serving request, a Transformer-based LLM operates through two phases: *Prefilling* and *Decoding*. During the prefilling phase, the entire user prompt is processed and the LLM generates the first output token. The user prompt can be processed in parallel in a single iteration. Therefore, GPU utilization is typically high during this prefilling phase thanks to the intra-request parallelism. During the decoding phase, the model generates output tokens sequentially, relying on all previously generated tokens, including both user prompts and all tokens produced thus far. This sequential generation inherently results in lower GPU utilization and throughput as only one token can be produced per iteration for a single serving request. This sequential generation process is called *autoregressive* decoding. Therefore, modern LLM serving systems batch multiple serving requests together to improve system throughput and resource utilization. For example, continuous batching Yu et al. (2022); Kwon et al. (2023) has become the most widely deployed batching technique in existing LLM serving systems.

## 2.2 LLM Serving with External Tool Invocations

Recently, there has been a rising trend of integrating LLM serving with external tool execution (e.g., ChatGPT plugins and Toolformer Schick et al. (2023)). External tools extend an LLM's capabilities to perform various complex tasks and interact with external environments. For example, many AI agents enable LLMs to access search engines to acquire up-to-date information or access databases that contain private datasets. LLMs can also invoke calculators to reason about complex math equations and execute code interpreters to generate customized data visualization. Moreover, they can trigger other ML models (e.g., computer vision models) to understand image contents.

A typical tool-enabled LLM serving system is depicted in Figure 2. A user sends a prompt to an intermediate agent. There are typically three components in such systems: an intermediate agent interacting with users, an LLM serving system, and a set of tool executors that interact with external environments. There can be several rounds of tool invocation when serving one user request. For brevity, we demonstrate the workflow assuming the request only needs one round of tool invocation: (a) the user first sends a request to the intermediate agent. (b) The agent feeds the original prompt to the LLM serving system, and (c) the LLM generates a plan, including tools to invoke and corresponding parameters. (d) The agent then invokes tool executors accordingly. (e) The tool executors interact with the external environments (*e.g.*, online search engines or databases) and (f) return the observations (*i.e.*, execution output) to the agent. (g) The agent concatenates the original prompt, plans, and observations together and feeds them back to the LLM. (h) The LLM generates the ultimate response and (i) the agent sends the response back to the user.

Integrating external tools into an LLM serving system has inspired a new line in machine learning system research: KV cache management. Since multiple rounds of LLM serving are typically needed for a single user request, the resources (*e.g.*, KV cache) for a finished LLM serving are likely to be reused by future LLM inferences Abhyankar et al. (2024). Treating these multiple rounds of LLM serving as independent requests results in redundant prefilling of the same token sequences. For example, in Figure 2, to process the serving request (7), the LLM needs to compute the KV vectors for the original prompt and generated plans, which have already been computed previously as (2) and (3). AttentionStore Gao et al. (2024) evicts the KV cache to CPU memory as long as the PCIe bandwidth permits the transfer. When the next round initiates, the KV cache is then loaded back into the GPU memory to eliminate KV cache recomputation. InferCept Abhyankar et al. (2024) predicts the execution time of external API access and estimates the corresponding GPU resource waste. InferCept then uses the estimation results to make GPU management decisions, such as discarding the key-value states, swapping the states to CPU memory, or retaining the states inside the GPU.

Managing the KV cache is only one of the opportunities to accelerate LLM serving with external tools shown in this of works. Next, we will show that there is one additional opportunity to improve LLM serving with external tool invocations.

## 3 DESIGN

We first describe the new opportunity of tool partial execution. We then describe our new tool interface design and system implementation in Conveyor. Finally, we analyze the potential performance gain from tool partial execution.

### 3.1 EFFICIENCY OPPORTUNITIES IN MODERN TOOL-ASSISTED LLM SERVING SYSTEMS

Today's tool-assisted LLM serving systems start the plan execution only after the LLM completes the entire decoding procedure. This misses the opportunities for pipelining LLM's decoding and plan execution to achieve reduced serving latency and improved system efficiency. For example, it would be ideal if the Python interpreter could start partially executing the script as soon as the LLM generates the first line of code (*e.g.*, `import torch`), without waiting for the generation of the entire script. However, in existing designs Schick et al. (2023); Abhyankar et al. (2024), since the LLM serving system and the tool execution are not co-optimized, the Python interpreter will only start after the entire script has been generated. This leads to extended serving delay and inefficient resource utilization (*e.g.*, the LLM is idle and GPU cycles are wasted). We name this desired capability of initiating tool execution before complete LLM decoding as *tool partial execution*, which has the potential to significantly reduce the serving latency and improve the overall efficiency and responsiveness of the system.

Let's consider a concrete example of executing Python code (in Figure 3). This code is generated by Mistral-7B-Instruct-v0.2 with the prompt "Plotting a sine wave in python with torch and matplotlib. ONLY output code without trailing explanation.". We use the markdown code block syntax ```` ```python ```` and ```` ``` ```` as the indicators for the start and end of the

```python
```python
import torch
import torchvision.utils as vutils
import matplotlib.pyplot as plt
import numpy as np

x = torch.linspace(0, 2 * torch.tensor(np.pi), 1000)
y = torch.sin(x)

vutils.save_image(y.unsqueeze(0), 'sin_wave.png')
plt.plot(x.numpy(), y.numpy())
plt.show()
```
```

Figure 3: Python code generated by the LLM.

tool. The execution without tool partial execution is that the LLM first generates the entire Python script, executes the script, and returns the image to the user. To understand why Conveyor provides performance improvement in this case, we plot the execution timeline with and without tool partial execution in Figure 4. The green blocks represent the LLM decoding of a line of Python. The numbers represent line numbers. The grey boxes represent the execution of the Python code in a Python interpreter. With partial execution, the execution of lines 1–12 can be completely pipelined with the decoding procedure, and only line 13 needs to be executed after the decoding is finished.

Realizing such partial execution in existing tool-assisted LLM serving systems requires us to address the following two technical challenges. First, the LLM system needs to understand *when a tool partial execution can be started*. This information needs to be passed to the system whenever a tool is registered, and it varies across different tools. For example, it is not possible to establish an HTTP connection without fully decoding the hostname. Therefore, a new set of interfaces should be properly designed for tool developers. Second, we need to carefully avoid unnecessary blocking and maximize resource efficiency when LLM decoding and tool execution are scheduled in parallel. For example, one round of LLM serving may invoke multiple tools sequentially. The system should manage these executions and outputs properly so that tool execution will not affect LLM decoding or vice versa.

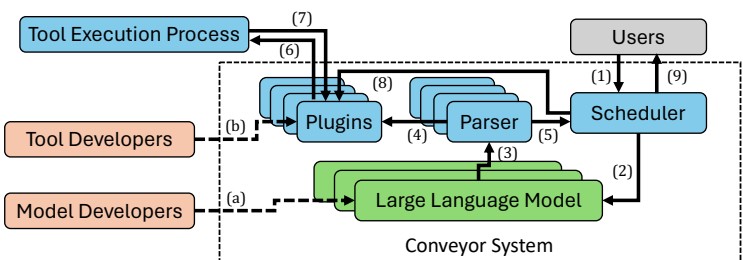

Figure 4: Case #1: Execution timeline for the CodeGen workload with and without partial execution. The numbers in the diagram represent the line number of code in Figure 3. The length of each block represents the relative execution time but does not correspond to exact duration due to the expressiveness constraints in the diagram.

Figure 5: Conveyor workflow overview.

## 3.2 TOOL INTERFACE DESIGN TAILORED FOR PARTIAL EXECUTION

Partial execution in Conveyor requires tool developers' involvement to achieve optimal performance. Tool developers need to inform the LLM system when a tool can be initiated and what data is needed for the tool to execute. One option is to provide a token-level streaming interface to tool developers. This option requires tool developers to manually parse these tokens and extract the required information (e.g., tool invocation indicator and corresponding parameters). However, this option is neither user-friendly nor efficient. First, the tool developer has to handle complex parsing logic based on raw tokens, which can be burdensome for complex tools. Furthermore, a user may register many tools for potential usage while an LLM request may invoke none of them. In this scenario, complex parsing can be redundantly executed multiple times because each tool needs to process raw tokens independently, leading to wasted resources.

Instead, Conveyor takes over the parsing responsibility and provides neat but generic interfaces to tool developers. Conveyor offers a set of parsers and a base plugin interface for tool developers. Tool developers only need to select a parser, typically depending on the LLM they use, and wrap their tool implementation with the plugin interface. Developers register both the parser and the plugin with the system during registration.

The parser and plugin interface enable the system to determine when a tool is invoked and how data is used by the tool (e.g., data format). For example, a Python interpreter plugin consumes data line by line as it executes a line of data when each end-of-line is decoded. Conveyor handles token parsing for all tools, avoiding redundant parsing execution by different tools. Additionally, wrapping the tool implementation with our plugin interface is lightweight. For example, our implementation for the Python interpreter plugin shows that only 32 lines of Python code are needed, including necessary logging and error handling.

## 3.3 CONVEYOR EXECUTION WORKFLOW

After plugins are registered to Conveyor, the system understands the indicators of different tools for partial execution and the data these tools require. To efficiently and effectively detect these indicators and conduct partial execution during decoding, Conveyor includes an efficient parser. This parser parses the generated token stream and invokes plugins accordingly. It is designed to support stream processing and emit completed pieces of data immediately for efficiency.

Figure 5 shows the system architecture and workflow of Conveyor. Model developers and tool developers first register models and tool plugins to Conveyor, shown as (a) and (b). Next, when

(1) a user sends a request to the system, the scheduler schedules received requests and (2) invokes the LLM for serving. During each decoding iteration, (3) tokens are sent to the parser. The parser processes generated tokens, assembles them into semantic information (e.g., strings or keys), and (4) identifies tool invocation indicators. When a tool invocation indicator is identified, (6) Conveyor invokes the corresponding plugins and spawns a new process with an isolated setup to run the tool instance. Conveyor relies on duplex inter-process communication (IPC) channels to communicate with the new process running the tools to (6) send tool execution commands and (7) receive outputs. If no tool invocation is needed, (5) the parser simply returns the tokens to the scheduler and the next iteration starts. During each iteration, (8) the scheduler periodically polls the plugins' status to receive tool execution outputs. The scheduler determines whether (2) a new round of serving is needed or (9) the response is ready to be returned to the user.

Currently, LLMs continue to decode the data needed by the tool after the indicator has been decoded. Therefore, when the parser has processed adequate tokens and detects that a piece of data needed by the tool has been decoded, it assembles a message containing the data and sends it through the IPC channel to the tool process. Notably, the parser only needs to wait for the data required by the tool (e.g., parameters) instead of the entire LLM response. The separate process receives the data from the IPC channel and attempts to execute the tools. Depending on the tools, there might be cases where a tool needs multiple pieces of data to execute. The process executing the tools will store data in an internal buffer and wait for the remaining data in such cases. For example, when invoking a Python interpreter, function definitions (e.g., `def func():`) should be executed when the entire definition block has been decoded.

It is worthwhile to note that we choose to spawn tool execution on a separate dedicated process to avoid unnecessary contention or interference with the LLM decoding procedure. Machine learning serving software usually uses Python as the programming language platform (e.g., PyTorch Paszke et al. (2019)), and running a Python-based tool in the same process may cause either the tool execution or LLM decoding to be blocked by the Python Global Interpreter Lock (GIL) mechanism. Such contention can cause extended latency and reduce system performance. Further, running tools on a separate process will enforce security, since the tools are run in a different context, having an isolated address space. Even if the tool is corrupted, it will not affect the integrity of the LLM inference. Another feature of Conveyor is that the parser implementation only depends on the syntax of the tool message generated by the model. This means Conveyor only rely on a small number of well-defined parsers. The number of parsers needed only depend on the number of models supported by the system, no the number of tools.

### 3.4 THEORETICAL ANALYSIS

We now analyze the theoretical performance gain brought by Conveyor. In §4.4, we demonstrate that the theoretical performance gain can reflect the general trend of empirical performance gain measured in our real implementation.

Consider a general tool-assisted LLM serving request, which may consist of multiple rounds of tool invocations. For each round $i$, we denote the time of token generation as $g_i$, including both the prefilling and decoding time. We denote the time of tool execution for round $i$ as $t_i$. In tool-assisted LLM serving workflows, the next generation (*i.e.*, decoding) phases typically depend on previous tool outputs. Therefore, let us assume that the generation of $g_{i+1}$ depends on the tool invocation of $t_i$. Consider $n$ rounds of tool invocations. Without tool partial execution, the total execution time is $L_{old} = \sum_{i=1}^{n}(g_i + t_i) + g_{n+1}$. Here $g_{n+1}$ is the LLM decoding to process the output of the last tool invocation, so there is no tool access after this decoding procedure.

When tool partial execution is enabled (*e.g.*, using Conveyor), the best case is that the token generation and tool execution can be fully parallelized. For example, the tool starts to execute after the first token is generated and returns the output before the last token is decoded. The worst case is that the tool only starts after the entire decoding procedure has finished. Therefore, the theoretical time consumed by each tool-assisted LLM serving round $i$ would be

$$\max\{g_i, t_i\} \leq L_i \leq g_i + t_i. \tag{1}$$

The overall latency to serve a request is therefore bounded by:

$$\sum_{i=1}^{n} \max\{g_i, t_i\} + g_{n+1} \leq L_{new} \leq L_{old} \tag{2}$$

The best-case speedup that can be achieved via enabling tool partial execution is where $g_i$ and $t_i$ are fully overlapped, *i.e.*, $L_{new} = \sum_{i=1}^{n} \max\{g_i, t_i\} + g_{n+1}$, with a corresponding relative latency improvement given by $\frac{\sum_{i=1}^{n}(g_i+t_i)+g_{n+1}}{\sum_{i=1}^{n} \max\{g_i,t_i\}+g_{n+1}} - 1$.

## 4 EVALUATION

We evaluate Conveyor on various workloads and demonstrate how integrating tool-awareness into modern LLM serving systems enhances system efficiency. First, we briefly introduce our evaluation setup. We then evaluate Conveyor on various tool-assisted LLM serving tasks from existing literature, showing that partial tool execution significantly reduces response delay and improves overall resource utilization. Second, through two case studies, we systematically break down performance improvements for these workloads in detail. We then demonstrate that Conveyor matches the trend of the theoretical best-case latency speedup. Finally, we analyze and demonstrate that Conveyor's overhead is negligible in modern tool-assisted LLM serving scenarios.

### 4.1 SETUP

We evaluate Conveyor on our testbed of servers with two Intel 10-core Xeon Gold 5215 CPUs (running at 2.5 GHz base frequency) and one NVIDIA GeForce RTX 3090 GPU. We implement our system on top of PyTorch Paszke et al. (2019) with FlashInfer CUDA kernels Ye et al. (2024). Our Conveyor system is implemented in about 2K lines of Python. Our baseline is the same code but with tool partial execution disabled, where tool invocation always happens after decoding to the end-of-sequence (EOS) token.

**Workloads.** To the best of our knowledge, there are unfortunately no publicly available realistic datasets of tool-assisted LLM serving scenarios. Although enabling LLMs to use tools is a very hot field in generative AI, prior works Li et al. (2023); Xu et al. (2023b) mainly test whether LLM can produce correct output to invoke the tools. The tool API interfaces are designed for testing LLMs' comprehension capabilities, and the tools only have mocked backend implementation which produce synthetic results. We cannot use such workloads to evaluate Conveyor, because we need tools to actually execute in order to perform latency evaluation. Instead, we systematically investigate existing literature Kim et al. (2023); Liu et al. (2024); Jin et al. (2024); Arora & Kambhampati (2023); Xu et al. (2023a); Ruan et al. (2023); Kuchnik et al. (2023); Li et al. (2023); Xu et al. (2023b) and construct four scenarios for our evaluation. We implement generic tool interfaces and the backend for the following four scenarios. In our evaluation, the tools execute real actions and make actual network requests to the Internet.

- **CodeGen**: We ask the LLM to plot a sine wave in Python with the `torch` and `matplotlib` library. Tool partial execution starts when Conveyor detects a complete line of Python code is decoded.

- **Search**: We ask the LLM to write a "Hello World" program in Python, C++ and Java consecutively, using tools to search online and use results from StackOverflow. Tool partial execution starts when Conveyor identifies the function name of the tool.

- **Planning**: We ask the LLM to search the market caps of Microsoft and Apple, and use a calculator to compute their ratio and output using a given formatting tool. The LLM generates a 4-stage plan involving tools: the first and second stages search on the Internet, the third stage is to invoke a calculator, and final stage is to use the format tool. Tool partial execution starts when a complete stage of the plan is generated.

- **Validation\***: We ask the LLM to generate a function call to get local news. However, the LLM fails to generate the correct location arguments even though it is prompted in the tool description. The local news API requires a city name and a state name. If arguments are not correctly presented in the correct format (e.g., missing the state name or the city name), the API call will fail. The

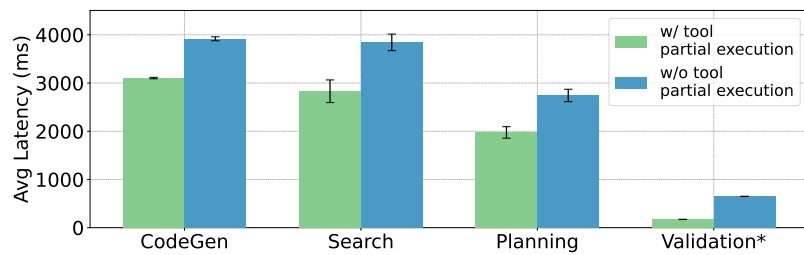

Figure 6: Average request completion latency with and without tool partial execution. Error bars represent the standard deviations. Validation*: validation is not a standalone tool; it is a functionality embedded within tools, such as verifying the validity of parameters.

    failure is due to the LLM model's reasoning constraint. For this workload, we only measure the latency needed for detecting if this invocation is problematic, not including tool execution time.

We use Mistral-7B-Instruct-v0.2 for Python CodeGen and Planning workloads, and use Functionary-Small-v2.2 for Search, Validation workloads. We choose Mistral-7B, because it is great at generating Python code. We pick Functionary-Small, because it is trained for invoking tools. At the same time, they meet our testbed's limitations (NVIDIA RTX 3090's available GPU memory). We pick Functionary-Small for the Validation workload because we empirically found that it has a higher probability of generating format-correct requests compared to Mistral-7B. We set temperature to be 0, so every test for the same workload has the same LLM output. Performance variance across tests for the same workload is due to the performance variance in CPU/GPU processing and the Internet (for the Search and Planning workloads).

Note that the effectiveness of Conveyor is not affected by the choice of LLM or prompts because the execution flow and of LLM decoding instructions and invoking tools remains the same. The quality of the final output will also not be affected since the output of both LLMs and tools are unmodified. Enabling Conveyor or not only affects the starting time of tool execution and thus improve latency. These aspects will be further elaborated in the following case study section.

### 4.2 MAIN RESULTS

Conveyor's performance improvement is significant, but the extent heavily depends on the performance characteristics of the tools. We run each workload 100 times and collect the average latency and the standard deviations, and the results are shown in Figure 6. For the CodeGen workload, the latency improvement comes from the parallelized execution of LLM decoding and the Python interpreter execution. The average latency improvement is 26.3%. For the Search workload, the performance gain comes from parallelizing the tool invocation of the search on StackOverflow and the LLM decoding for the next search. The improvement is 35.8% on average. The latency variance of the Search workload is high because it involves search over the Internet, and the constraints in the search bring more uncertainty at the server side. For the Planning workload, the improvement comes from parallel execution of decoding the plan and executing parts of the partially decoded plan, and the corresponding latency improvement is 38.8%. Validation workload has shown the best latency improvement of 376.4%, in which the source of the performance improvement is different from other workloads. The partial execution allows the check for the format of the tool execution to run before the entire sequence is decoded.

### 4.3 CASE STUDIES

Now, we delve deep into two case studies, CodeGen and Validation, to demonstrate where the performance improvement comes from.

**Case #1: Python code generation.** For the CodeGen workload, we let Mistral-7B-Instruct-v0.2 generates the Python code to plot a sine wave (Figure 3). Figure 4 shows the corresponding execution timelines with and without tool partial execution. Conveyor helps to reduce averagely *725 ms* for the entire end-to-end serving latency, which is 3,918 ms on average without partial execution, leading

Decoding — "location": "New York" — ✗ → Abort
Plugin w/ Validation
**With Partial Execution**

Decoding — "location": "New York", "date": "2024-04-01", "duration": … — ✗ → Abort
Plugin w/ Validation **Without Partial Execution**

■ Decoding ■ Prefilling ■ Tool Execution

Figure 7: Case #2: Execution timeline for the Validation workload with and without partial execution.

to *26.3%* improvement (see §3.4) for this user request. It is worthwhile to note that a few import statements (*e.g.*, line 2 and line 5) consume negligible time because Conveyor invokes the Python interpreter through `fork`, so the plotting script can reuse the modules already imported by Conveyor.

**Case #2: Tool validation.** For the Validation workload, we let Functionary-Small-v2.2 generate a request for a local news service. The request is in JSON format, and the field "location" has to contain a valid city name and a state name. If the "location" only contains the city name, the request will be rejected by the service, and there is no point in sending such a request. Figure 7 shows the execution timeline. Without partial execution, such a check has to take place after the entire request is decoded. Partial execution allows the check to happen earlier and can abort immediately, saving the resources and the time for decoding subsequent tokens.

## 4.4 LIMITATIONS OF CONVEYOR AND THEORETICAL ANALYSIS

The performance improvement of Conveyor depends heavily on the workloads. For lightweight tools (where tool execution time is orders-of-magnitude less than the decoding time), we would expect Conveyor to provide minimal performance improvement. To study this effect, we study two additional workloads, where there is negligible opportunity to overlap LLM decoding and tool execution. We use Functionary-Small-v2.2 for these two tools.

- **Database**: We provide a small SQLite file on disk in advance and ask the LLM to select all the data from the database. Tool partial execution starts when Conveyor identifies the function name of the tool.

- **Calculator**: We ask the LLM to compute $200 \times 701$ using a calculator tool. Tool partial execution starts when the complete formula is decoded.

Further, our system evaluation is limited to the capability of existing open-source models and the existing sets of tools that they support. Even for the tools evaluated in this paper, it is difficult to comprehensively evaluate more workloads due to model restrictions. For example, for the planning workloads, we are only able to evaluate plans that existing models can create, and future LLMs may be able to create more complex plans that our evaluation methodology cannot cover.

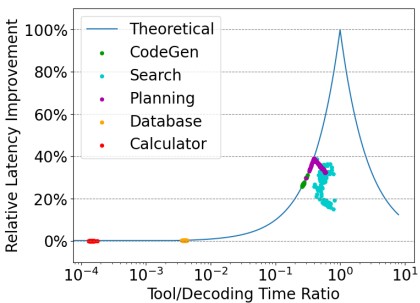

Figure 8: Theoretical and empirical latency improvement from tool partial execution.

To systematically quantify how much benefits Conveyor's tool partial execution can bring for workloads have only have little overlapping opportunity for tool execution and LLM decoding and future tools, we use our mathematical analysis in §3.4 to quantify the maximum theoretical performance gain in terms of latency improvement. We plot this latency improvement as a blue curve in Figure 8. The x-axis is the ratio between tool execution time and decoding time, $\frac{t_i}{g_i}$, assuming this ratio is fixed across all $i$ and $g_{n+1}$ is negligible compared to $\sum_{i=1}^{n} \max\{g_i, t_i\}$. We run empirical experiments for the Database workloads and Calculator workloads. We put our empirical evaluation results as red dots in the figure. As expected, the Database and the Calculator workloads achieve a near-zero improvement. This is because the tool execution time is negligible compared to LLM decoding time. They will not be accelerated by tool partial execution. The CodeGen, the Planning, and the Search workloads achieve a more significant performance improvement, because their tool/decoding time

ratio is near the peak of the curve. The red dots are below the theoretical limit, because the theoretical limit assumes tool access time is fully masked by decoding or vice versa. The red dots match the overall trend of the theoretical best-case latency speedup. Note that the Search and the Planning workloads performance depend on Internet performance. Their tool time is not stable and thus hard to overlap perfectly with LLM decoding. We did not show the dot for the Validation workload: its source of improvement is from aborting decoding earlier and is thus not captured by our theoretical analysis.

### 4.5 OVERHEAD OF CONVEYOR

Conveyor has overheads in parsing output tokens. We measure Conveyor's CPU overheads when running the CodeGen task with and without tool partial execution. Our result is that Conveyor incurs 0.6% extra CPU cycles, which is negligible. This is expected since most CPU cycles are used in the LLM serving (*e.g.*, launching CUDA kernels) and triggering external tools), and parsing itself is much more lightweight compared to these operations.

Another type of overhead is how much additional human effort a tool developer needs in order to use our interface to port tools on top of Conveyor. For our six workloads, we manually incorporate corresponding tools to Conveyor using Conveyor's tool plugin interface. Each tool's incorporation only needs 20–40 lines of code. This demonstrates that porting more tools on top of Conveyor will be simple.

## 5 CONCLUSION

In this paper, we presented Conveyor, a novel LLM serving system designed to efficiently handle requests that incorporate external tools. The core idea of Conveyor is to enable tool partial execution alongside LLM decoding to improve request completion latency. Conveyor's design consists of two components. First, Conveyor contains a tool interface design for tools to indicate the partial execution opportunity to an LLM serving system. Second, Conveyor has a request scheduler that facilitates corresponding tool partial execution. Our evaluation based on a set of LLM serving workloads shows that Conveyor improves request completion time by up to 38.8%.

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
