# OpenReview forum: "Conveyor: Efficient Tool-aware LLM Serving with Tool Partial Execution"
_ICLR.cc/2025/Conference — ICLR 2025 Conference Withdrawn Submission_

### Official Review · Reviewer_ZQMh · 2024-10-24

**Soundness:** 2
**Presentation:** 3
**Contribution:** 2
**Rating:** 5
**Confidence:** 4

**Summary:**

The authors present Conveyor, an approach that enables partial inference request processing with time delay considerations.

**Strengths:**

The paper touches on a very timely and important matter as inference optimization becomes increasingly important with broader adoption.
The paper has the following strengths:
- The experimental pipelines are well-chosen as I think they represent a good range of practical use cases.
- The theoretical framework is intuitive.

**Weaknesses:**

Score-relevant weaknesses:
- Are the partial execution triggers learned or rule-based? Things like a newline are straightforward, but what about specific details like code delimiters that vary across programming languages? I understand it has to be passed with a tool, but isn't it impractical to define potentially 100s or 1000s of triggers? Wouldn't learning be more appropriate, especially since you already have the tokens available? I would appreciate more details and a more thorough evaluation of these triggers.
- While Conveyor enables tool execution based on partial inference outputs, how does the scheduler compare to dynamic batching? How much performance (time, hardware utilization) does Conveyor gain compared to dynamic or continuous batching?
- In equation 2, What is the difference between $\sum_i^n\max(g_i, t_i) + g_{n+1}$ and $L_{new}$? By the authors' definition, they are identical in the best case. What is the worst-case assumption, or what is the penalty for inefficiency? I am missing the "optimization" criterion and how the Conveyor latency can be bounded lower than in sequential execution. Sure, parallelization helps, but it is trivial. I think a more thorough theoretical definition of partial execution triggers is needed.
- I would have appreciated an appendix with more experimental details since the work appears largely based on empirical evaluations.

Minor remarks:
- The way papers are cited appears strange. There are never brackets around the citations. This makes the paper hard to read.

**Questions:**

Please see the weaknesses.

---

### Official Review · Reviewer_6Uia · 2024-10-31

**Soundness:** 2
**Presentation:** 3
**Contribution:** 2
**Rating:** 3
**Confidence:** 4

**Summary:**

This paper presents Conveyor, an efficient LLM-serving system optimized for handling requests that involve external tools. It integrates tool partial execution with LLM decoding, demonstrating across four workloads that this approach reduces request completion latency.

**Strengths:**

(1) The writing of the paper is good.

(2) This paper proposes a method addressing a problem for which satisfactory solutions are currently lacking and offers a reference for future research.

**Weaknesses:**

(1) The related work is insufficient and does not demonstrate the advantages and differences of this work over prior studies. In the related work section(L94), the paper lacks an introduction to studies where researchers recognize methods for improving the efficiency of LLM external tool utilization such as LLM-dCache[1] and APIServe[2].

(2) The author’s approach lacks innovation and appears rather straightforward. Moreover, the effectiveness of this method may be highly dependent on the specific query and execution paradigms of the agent, with limited generalizability and applicability. The system utilizes a parser to schedule tasks based on predefined rules and existing tools. This paradigm may lack the capability to incorporate other information like feedback for improving scheduling strategies, making it poorly adaptable to varied inputs. In fact, a vast array of paradigms has been proposed for agent-based execution; for example, "React"[3] suggests concurrent feedback and execution will enhance performance. The author attempts to build a system based on a paradigm that is neither widely accepted nor the most effective. Additionally, the author does not demonstrate in experiments how their method integrates with various agent enhancement techniques, such as caching and memory, which further limits the significance of this work.

(3) The experiments presented in this paper are insufficient. In the experimental section, the authors propose to validate the effectiveness of their method across four workloads; however, they conducted only a single experiment for each workload. The lack of multiple experimental trials renders the results less representative and fails to demonstrate that the proposed method can be universally applicable to other tasks within the same workload.


[1] Singh S, Fore M, Karatzas A, et al. LLM-dCache: Improving Tool-Augmented LLMs with GPT-Driven Localized Data Caching[J]. arXiv preprint arXiv:2406.06799, 2024.
[2] Abhyankar, Reyna, Zijian He, Vikranth Srivatsa, Hao Zhang, and Yiying Zhang. "APIServe: Efficient API Support for Large-Language Model Inferencing." arXiv preprint arXiv:2402.01869 (2024).
[3]Yao S, Zhao J, Yu D, et al. React: Synergizing reasoning and acting in language models[J]. arXiv preprint arXiv:2210.03629, 2022.

**Questions:**

(1) Does your method demonstrate robust performance across other tasks within the same workload? For instance, in a code generation workload, can the method effectively reduce the overall execution time when generating more complex code?

(2) Are there any specific limitations or boundary conditions? Please clarify how these limitations may affect the application of your method.

(3) Could you further expound on the distinctions and advantages of this method compared to others?

---

### Official Review · Reviewer_rZ5t · 2024-11-03

**Soundness:** 2
**Presentation:** 3
**Contribution:** 2
**Rating:** 3
**Confidence:** 4

**Summary:**

The paper introduces Conveyor, an optimized LLM system augmented with external tools to improve latency by enabling partial execution of these tools during LLM decoding. Conveyor is built on token-granularity scheduling and includes an interface that allows tool developers to specify when partial execution can start, facilitating concurrent LLM decoding and tool execution.

**Strengths:**

-The paper tackles challenges associated with augmented LLMs, advancing the development of compound AI systems.

-The paper provides a comprehensive breakdown of the workflow for LLMs with external tool augmentation, thoroughly explaining each design component.

-Evaluation covers diverse workloads—code generation, search, planning, and validation—demonstrating Conveyor’s performance across various scenarios.

**Weaknesses:**

- The impact of the contribution is limited by its reliance on specific types of external tool calls and workload characteristics. The optimization benefits only long, independent tool calls, raising questions about its broad applicability. Additionally, the paper does not rigorously analyze the potential decoding overhead.

- Conveyor could potentially increase latency in cases where its overhead outweighs the benefits. Presenting these cases would add value, and a hybrid approach that dynamically enables or disables the optimization based on predicted tool and system properties could be more effective.

- The system’s ability to recognize when partial execution can start would require adaptation with each new capability, limiting generalization.

- Section 3.3 describes Conveyor’s parser as “efficient,” but more clarity and specific metrics would help substantiate this claim.

- The theoretical analysis omits the overhead associated with token passing and does not account for the likelihood of dependencies that could delay tool execution.

- The evaluation should incorporate state-of-the-art external tool augmentation methods (INFERCEPT) as a comparison baseline.

- Although the paper notes the lack of realistic datasets for tool-assisted LLM scenarios, ToolBench includes data for external tool augmentation and could be a valuable addition.

- The number of code lines may not accurately reflect human effort, as complexity and adaptability also impact implementation ease.

**Questions:**

- How does Conveyor align with efforts to minimize GPU waste in LLM systems (INFERCEPT)?
- Could you clarify the extent to which Conveyor’s hybrid approach might be feasible, allowing dynamic adjustment of partial execution based on tool or workload characteristics?

---

### Official Review · Reviewer_dPxt · 2024-11-06

**Soundness:** 1
**Presentation:** 2
**Contribution:** 1
**Rating:** 3
**Confidence:** 3

**Summary:**

The paper introduces Conveyor, an efficient LLM serving system optimized for the latency of workloads involving tool executions. Conveyor achieves this by separating text generation from tool execution and running them in parallel. The authors design parsers within the prompting interface to identify tool execution commands. They evaluate their approach to various tool execution tasks, demonstrating that parallel execution of text generation and tool execution significantly reduces latency compared to sequential execution.

**Strengths:**

The concept of separating text generation from tool execution and running them in parallel is interesting.
The background introduction to the key concepts in LLM serving and the tool execution workflow is correct.
The paper involves some engineering effort in prompt design.

**Weaknesses:**

The paper makes a strong and somewhat unrealistic assumption. Based on the illustrative examples (Figure 4), theoretical analysis (Section 3.4), and evaluation (Section 4), it seems the authors implicitly assume that each request triggers only one tool execution and does so only once. This oversimplification deviates significantly from real-world workloads.

In Section 3.4, the authors provide theoretical lower and upper bounds for their proposed parallel scheduling approach. However, these bounds are not particularly useful due to the strong implicit assumption and the fact that the resulting bounds still remain quite loose. The analysis assumes a single tool call per inference request, in which case the latency of parallel execution of decoding and tool execution falls in the range of [the duration of the longer task (either decoding or tool execution), latency of sequential execution of decoding + task]. But this offers little insight, as the range of improvement is too broad and lacks meaningful quantification.

The writing is imprecise and somewhat misleading. The so-called “serving system” is, actually, just a prompting interface. It does not address key challenges typically associated with optimizing serving systems, such as improvements at the model, algorithm, or system level. Similarly, the “workloads” are simplified use cases, which fail to capture the statistical characteristics of real-world workloads. The “parallel execution” described appears to merely separate text generation and tool execution into distinct prompt calls. In standard terminology, “parallel” usually implies the use of multi-threading, multi-processing, or hardware-level optimizations.

Formalizing the problem with accurate definitions of decoding, tool execution, timeline, and pipeline, and implementing the proposed solution in a real serving system would make the paper a stronger case.

**Questions:**

The authors call Conveyor a system, but I do not see system implementations except for a parser in prompting LLMs. Can you implement this into the current mainstream serving system, such as vLLM?
How would the results change if we could invoke different tools or invoke tools multiple times per request?

---

### Note · Authors · 2024-11-20

I have read and agree with the venue's withdrawal policy on behalf of myself and my co-authors.